# Physiological changes in retinal layers thicknesses measured with swept source optical coherence tomography

Elisa Viladés[1,2], Amaya Pérez-del Palomar[2,3], José Cegoñino[2,3], Javier Obis[1], María Satue[1,2], Elvira Orduna [1,2], Luis E. Pablo[1,2], Marta Ciprés[1,2], Elena Garcia-Martin [1,2] *

1 Department of Ophthalmology, Miguel Servet University Hospital, Zaragoza, Spain, 2 Aragon Institute for Health Research (IIS Aragón), University of Zaragoza, Zaragoza, Spain, 3 Mechanical Engineering Department, Aragon Institute of Engineering Research, University of Zaragoza, Zaragoza, Spain

* egmvivax@yahoo.com

**Data Availability Statement:** All relevant data are within the paper and its Supporting Information files.

## Abstract

### Purpose

To evaluate the physiological changes related with age of all retinal layers thickness measurements in macular and peripapillary areas in healthy eyes.

### Methods

Wide protocol scan (with a field of view of 12x9 cm) from Triton SS-OCT instrument (Topcon Corporation, Japan) was performed 463 heathy eyes from 463 healthy controls. This protocol allows to measure the thickness of the following layers: Retina, Retinal nerve fiber layer (RNFL), Ganglion cell layer (GCL +), GCL++ and choroid. In those layers, mean thickness was compared in four groups of ages: Group 1 (71 healthy subjects aged between 20 and 34 years); Group 2 (65 individuals aged 35–49 years), Group 3 (230 healthy controls aged 50–64 years) and Group 4 (97 healthy subjects aged 65–79 years).

### Results

The most significant thinning of all retinal layers occurs particularly in the transition from group 2 to group 3, especially in temporal superior quadrant at RNFL, GCL++ and retinal layers (p≤0.001), and temporal superior, temporal inferior, and temporal half in choroid layer (p<0.001). Curiously group 2 when compared with group 1 presents a significant thickening of RNFL in temporal superior quadrant (p = 0.001), inferior (p<0.001) and temporal (p = 0.001) halves, and also in nasal half in choroid layer (p = 0.001).

### Conclusions

Excepting the RNFL, which shows a thickening until the third decade of life, the rest of the layers seem to have a physiological progressive thinning.

**Funding:** EGM PI17/01726 (https://www.isciii.es/
QueHacemos/Financiacion/Paginas/default.aspx)
The funders had no role in study design, data
collection and analysis, decision to publish, or
preparation of the manuscript. EGM MAT2017-
83858-C2-2 MINECO/AEI/FEDER, UE, DPI2016-
79302R (https://www.isciii.es/QueHacemos/
Financiacion/Paginas/default.aspx) The funders
had no role in study design, data collection and
analysis, decision to publish, or preparation of the
manuscript.

**Competing interests:** The authors have declared
that no competing interests exist.

## Introduction

Currently OCT is widely used in clinical practice and clinical trials accepting their measurements for the evaluation of the response to treatment and the progression of pathologies [1]. Retinal thickness or central macular thickness (CMT) measured with OCT is particularly used, which correlates with pathological changes and response to treatment for a variety of eye diseases [2].

Currently we still accept the thickness of the retina as the space between surfaces detected, but retinal image segmentation is challenging; structures such as vascular structures, macula, and microaneurysms have low contrast with their background. In contrast, other structures have high contrast with background tissues, but they are difficult to distinguish using classical segmentation techniques [1, 3].

Also, recent works have shown that the most commonly used algorithms in daily practice for retinal layer segmentation have a large number of segmentation errors, especially in the case of age-related macular degeneration, great disruptive pathology such as subretinal fluid, intraretinal cysts and retinal detachments that interrupt the structured logical organization of the retinal layers [4, 5].

Currently we can classify the existing segmentation algorithms into two clusters, mathematical modeling and machine learning approaches. Mathematical modeling is based on the previous anatomical, structural and clinical knowledge that is known about the retina. However pure machine learning algorithms for retinal layer segmentation classifies each pixel from an image on how they fall under a particular layer or boundary, that means that boundaries between layers are not linear [2].

Swept-source (SS)-OCT offers potential advantages due to a modified Spectral-domain (SD) and depth resolved technology which includes an improved imaging range, minor sensitivity roll-off with imaging depth, greater detection efficiencies, and an adjustability to longer imaging wavelengths of 1050nm, this allows a greater choroidal penetration and higher speed for image acquisition. The main difference of SS-OCT is that captures the interferences of the backscattered light from the retina thanks to a wavelength sweeping laser light source and a photodiode detector, in contrast to SD-OCT where a line scan camera and a spectrometer record the interferences between a broadband light source. Those improvements on SS-OCT enables higher density raster scan protocols and deeper image penetration, as a result on en-face reconstructions a better visualization of choroidal detail is possible [6].

Swept-source (SS)-OCT with a wavelength of 1,050 nm and 100,000 A-scans/sec has allowed in-depth visualization of the eye from the retina to the sclera even in patients with moderate to severe cataracts, as well as during eye blinking and/or ocular movement. SS-OCT systems have the potential for superior and simultaneous imaging of the retina and choroid because of the longer wavelength, potentially higher detection efficiency, and lower dispersion [7].

Choroidal thinning has been considered more and more important in the last months, because it has been associated with some ophthalmological pathologies such as age-related macular degeneration (AMD), and also in neurodegenerative diseases such as multiple sclerosis [8], Parkinson's disease [9] or Fibromyalgia [10] and systemic conditions like diabetes mellitus [11] or pathologies with unclear physiopathology such as migraine [12, 13].

Also, age has been found to be negatively correlated with central choroidal thickness and with central choroidal volume [14], choroidal thickness and volume are also negatively statistically significant concerning the refractive error, and axial length measured with low-coherence reflectometry was also found to be negatively correlated with choroidal thickness and volume. On the other hand, sex has not been found to influence choroidal thickness significantly [15].

Imaging of the choroid was dramatically improved with the development of spectral domain optical coherence tomography (SD-OCT) and was further augmented with the advantage of enhanced depth imaging SD-OCT (EDI SD-OCT) by Spaide and colleagues [16].

Even though, standard cross-sectional (Bscan) OCT imaging are still limited, consequently the choroidal assessment is not as detailed as it could be. However, en-face OCT imaging, is able to provide a high-definition three-dimensional and depth-resolved reconstruction of the choroid, revealing choroidal vascular details not easily visible on cross-sectional OCT imaging. Despite this, SD-OCT is the gold standard for clinical assessment and management of chorioretinal disorders, nevertheless the limited depth of penetration (~850nm), could compromise the choroidal assessment although the selection of the enhanced depth imaging (EDI) method [7].

In the absence of automated segmentation software for SS OCT systems, previous investigators have used manual (mostly single-point) measurement techniques using in-built calibers or modification of retinal segmentation lines to evaluate choroidal thickness; given the high anatomic variability of the choroid, these are impractical for clinical use, are highly dependent on location of measurement, and may be subject to further operator effects [17]. Nevertheless, SS-OCT provides objective and automated measurements of the choroid.

## Materials and methods

Between 2015 and 2018, 480 healthy withe caucasian patients were recruited for this cross-sectional study. The inclusion criteria were age between 20 and 79 years, refractive error less than ± 5 diopters (D), axial length between 21 and 25 mm, intraocular pressure less than 21mmHg. The exclusion criteria were concomitant ocular disease (such as glaucoma or retinal pathology); systemic pathologies that could impair the visual system; ocular trauma; laser therapy; and pathology affecting the optic nerve and retina (such as glaucoma, optic neuritis, macular degeneration). We excluded eyes with morphometric parameters of optic disc suggestive of subclinical chronic glaucoma (cup to disc ration≥0.5).

All participants provided written informed consent to participate in this study. The written informed consent for participants and for the study protocol was approved by the Ethics Committee of Clinic Research in Aragon (CEICA) and by the Ethics Committee of Miguel Servet University Hospital, in Zaragoza, Spain, which specifically approved the study procedures. This study was conducted in accordance with the guidelines established by the principles of the Declaration of Helsinki.

Both eyes of each subject were evaluated, but only one of the eyes, randomly selected, was included in the statistical analysis to avoid potential bias by interrelation between eyes of the same subject. In the cases of subjects with exclusion criteria in only one eye, the other eye was selected for the analysis.

A total of 13 eyes were excluded because of not enough OCT quality or exclusion criteria detected during the exploration (morphometric parameters of optic disc compatibles with glaucoma, epiretinal membrane or macular hole). Finally, we included 467 eyes (233 males, 234 females), that were classified in four groups in steps of 15 years, as the difference between the youngest and the oldest is 60 years: Group 1 (composed by 71 eyes of 71 healthy subjects aged between 20 and 34 years); Group 2 (composed by 65 eyes of 65 healthy individual aged 35 to 49 years), Group 3 (with 230 eyes from 230 healthy controls aged between 50 and 64 years) and Group 4 (composed by 97 eyes from 97 healthy subjects aged 65–79 years).

### OCT evaluation

Retinal measurements were obtained in all subjects using the DRI Triton SS-OCT device (Topcon, Tokyo, Japan). We performed the 3D+5LineCross protocol (12.0*9.0mm + 9.00mm

overlap 8), which allows 100.000 A scans/sec. The DRI Triton SS-OCT provides a quality scale in the image to indicate the signal strength. The quality score ranges from 0 (poor quality) to 100 (excellent quality). Only images with a score >55 were analyzed in our study, and poor-quality images prior to data analysis were rejected.

We exported data of 5 layers using 3DH_DISC (Disc5.2x5.2-Superpixelgrid-200) and the thickness of the 26x26 grid were analyzed to get mean and standard deviation in the four age groups as shown in Fig 1.

Using the Data Collector software of the Triton OCT, measurements of full layers and of 5 different layers were obtained (Fig 2): Retinal thickness (from the inner limiting membrane–ILM- to the retinal pigment epithelium boundaries), Retinal nerve fiber layer (RNFL) (between the ILM to the GCL boundaries), Ganglion cell layer (GCL) + (between RNFL to the inner nuclear layer boundaries) and GCL++ (between ILM to the inner nuclear layer boundaries), and choroid (from the Bruch membrane to the choroidal-scleral interface).

Automated built-in calibration software, Topcon Advances Boundary Software (TABS) determined the distance between the delimiting lines in retina and choroidal plexus, establishing 7 boundaries and five layers.

## Data analysis

Comparison between age groups was performed using analysis of variance (ANOVA) for each thickness measurement for four quadrants and four halves. Post-Hoc analysis was performed to obtain statistical differences in each comparison between the four age groups (Group 1 compared with Group 2; Group 1 compared with Group 3; Group 1 compared with Group 4; Group 2 compared with Group 3; Group 2 compared with Group 4; and Group 3 compared with Group 4). Correction for multiple comparisons was used in all analyses. In order to graphically see the evolution of the thickness of each segmented layer, a numerical method called the finite element method (FEM) was used. Numerical techniques such as the finite

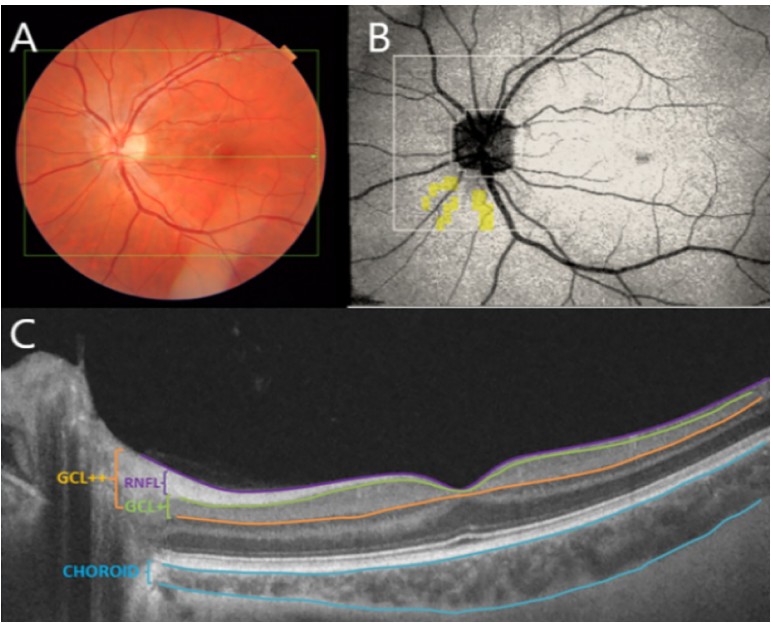

**Fig 1.** A: Location of 3D+5line cross OCT scans on retina. B: 26*26 grid centered on optic disc, the center area is not shown because the OCT thickness is always zero. C: cross-sectional OCT image and segmentation boundaries.

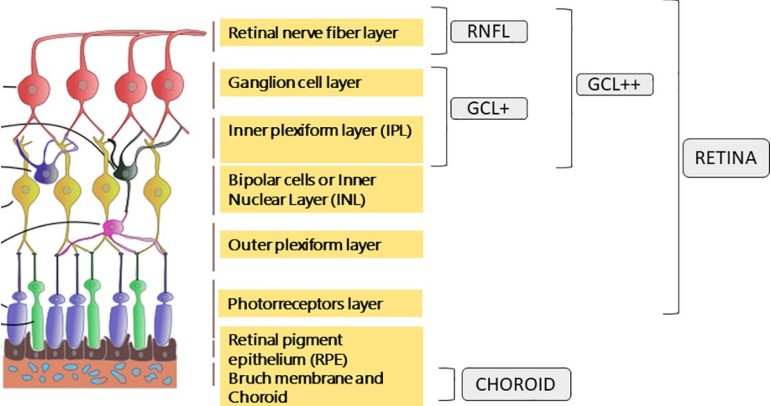

**Fig 2. Representation of the five layers measured by Triton optical coherence tomography.** Retina (from the inner limiting membrane–ILM- to the retinal pigment epithelium boundaries), Retinal nerve fiber layer (RNFL) (between the ILM to the GCL boundaries), Ganglion cell layer (GCL) + (between RNFL to the inner nuclear layer boundaries), GCL++ (between ILM to the inner nuclear layer boundaries), and choroid (from the Bruch membrane to the choroidal-scleral interface).

element method have been extensively implemented as effective and noninvasive methods to analyze biological tissues, and in particular in ophthalmology [18–20]. The finite element method is a computational tool which allows analysing the stress/strain behaviour of a structure subjected to different loads and boundary conditions. Here, this method has only been used to study the evolution of retinal layers thickness along time and to graphically see the evolution of each of them by a contour plot map. Thus, a finite element mesh using ABAQUS (Abaqus 6.14, Simulia, Rhode Island, USA) mimicking the Triton grid (Disc5.2x5.2-Superpixelgrid-200) was developed. The finite element mesh was constructed using membrane quadrilateral elements and the size of the mesh was 5.2x5.2mm. Afterwards, the thickness of each box of the grid for each group and each layer was introduced. In this way, the evolution of the thickness of each layer can be seen from a spatio-temporal point of view from Group 1 to Group 4.

## Results

We analysed a total of 467 eyes from 467 healthy subjects, 71 eyes from 71 individuals between 20–34 years (group 1), 65 eyes from 65 subjects between 35–49 years (group 2), 230 eyes from 230 subjects between 50–64 years (group 3) and 97 eyes from 97 individuals between 65–79 years (group 4).

Mean and standard deviation for each layer were represented in Fig 3 for the four age groups.

Fig 4 represents the contour map obtained by finite element modeling of the evolution of the thickness of each layer for the different age groups, which shows progressive thinning of all layers with age, especially from 50 years onwards.

A comparative analysis between age groups were performed for all layers to find thickness differences for four quadrants (temporal superior -TS-; temporal inferior -TI-; nasal superior -NS-; and nasal inferior -NI-) and four halves (superior, inferior, nasal and temporal).

### Retinal full layer analysis

There are highly significant differences between groups in our analysis for each quadrant and halves. In the post-hoc analysis, a significant thickening that did not overcome the Bonferroni

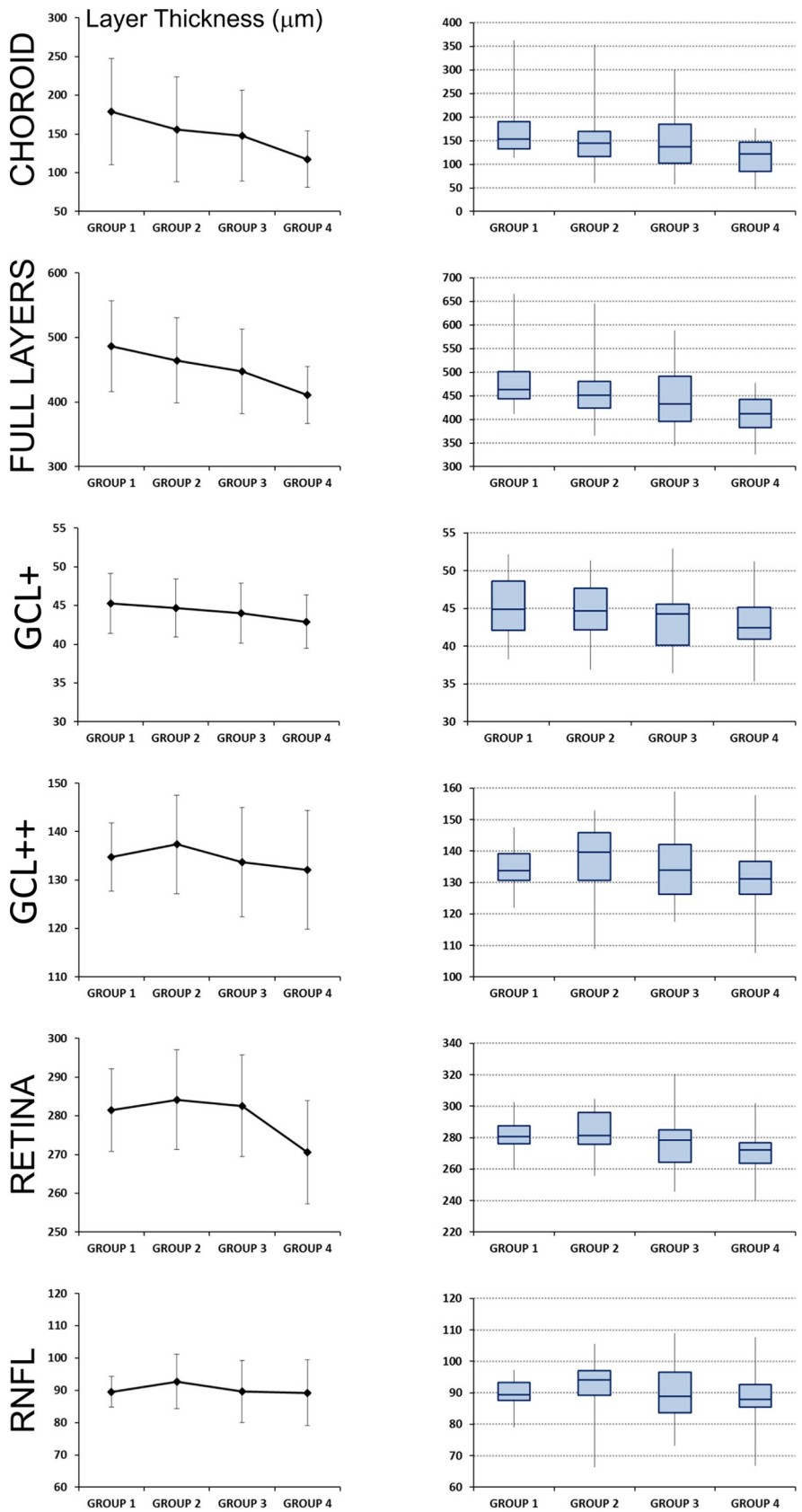

**Fig 3. Representation of mean and standard deviation for each layer measured by Triton optical coherence tomography, for the four age groups.** Group 1 (composed by 71 healthy subjects with age between 20 and 34 years); Group 2 (composed by 65 individuals with 35–49 years), Group 3 (with 230 healthy controls with 50–64 years) and Group 4 (composed by 97 healthy subjects with 65–79 years). On the left column, the mean and standard deviation is shown for each layer and age group. On the right column, the quartiles are plotted to show data dispersion and the presence of outliers.

correction for multiple comparison was found for TI quadrant at group 2 compared with group 1 (p = 0.038) and also at temporal half (p = 0.043). Considerable differences that overcame the Bonferroni correction or multiple comparison were found comparing group 2 vs group 3, in which a thinning trend is visible for TS (p = 0.004), and TI (p = 0.007) quadrants and temporal (p = 0.005) halves. Comparing group 3 vs group 4, significant thinning was found for the four quadrants (p≤0.002), and the four halves (p<0.001). Comparing group 1 vs group 3 we did not find any statistical differences between groups although there is a presumed thinning in group 3. On the other hand, there is an obvious thinning in group 4 compared with group 1 and group 2 for all quadrants and halves (p<0.001).

### Retinal nerve fiber layer analysis

Significant differences for all quadrants and halves were found between age groups, except for NS quadrant and nasal half. A significant RNFL thickening that overcame Bonferroni correction was observed at group 2 compared with group 1 in TS quadrant (p = 0.001), inferior half (p<0.001) and temporal half (p = 0.001). Group 3 presents a significant thickening at inferior half (p = 0.002) when compared with group 1, but a significant thinning at TS quadrant (p = 0.001) when compared with group 2. Group 4 presents just a significant thickening at NI quadrant (p = 0.002) compared with group 1, but compared with group 2, it is observed a significant thinning at TS (p<0.001) and TI (p<0.001) quadrants and at superior (p<0.001) and temporal (p<0.001) halves. Same conduct happens when this group is compared with group 3 at TI quadrant (p<0.001) (Table 1).

### RNFL to inner nuclear layer (GCL+)

GCL+ appears to be significantly thicker in group 2 when comparing with group 4 at TS quadrant (p = 0.001), inferior and temporal halves (p = 0.001), but not compared with groups 1 and 3. Group 3 present a significant thinner GCL+ layer at NI quadrant and inferior half (p<0.001) when compared with group 1, however GCL+ is significantly thicker in this group compared with older population (group 4) at TS quadrant and superior half (p = 0.001). Finally, group 4 exhibits a significant thinning of GCL+ layer for every quadrants and halves (p<0.001) (Table 1).

### From inner limiting membrane to inner nuclear layer (GCL++)

Significant differences between groups in the GCL++ layer were found almost in every quadrant and half, except for NI quadrant and nasal half, that apparently are not predictive of changes in this layer (Table 1). Comparing group 3 vs group 1, significant differences were not found, but compared with group 2 significant thinning appears in TS quadrant (p<0.001). On the other hand, group 4 exhibits a thinner GCL++ layer compared with group 1 in TI quadrant (p = 0.001) and superior and temporal halves (p = 0.002). Compared with group 2 thinning gets more significant in TS and TI quadrants (p<0.001), and superior and temporal halves (p<0.001) (Table 1).

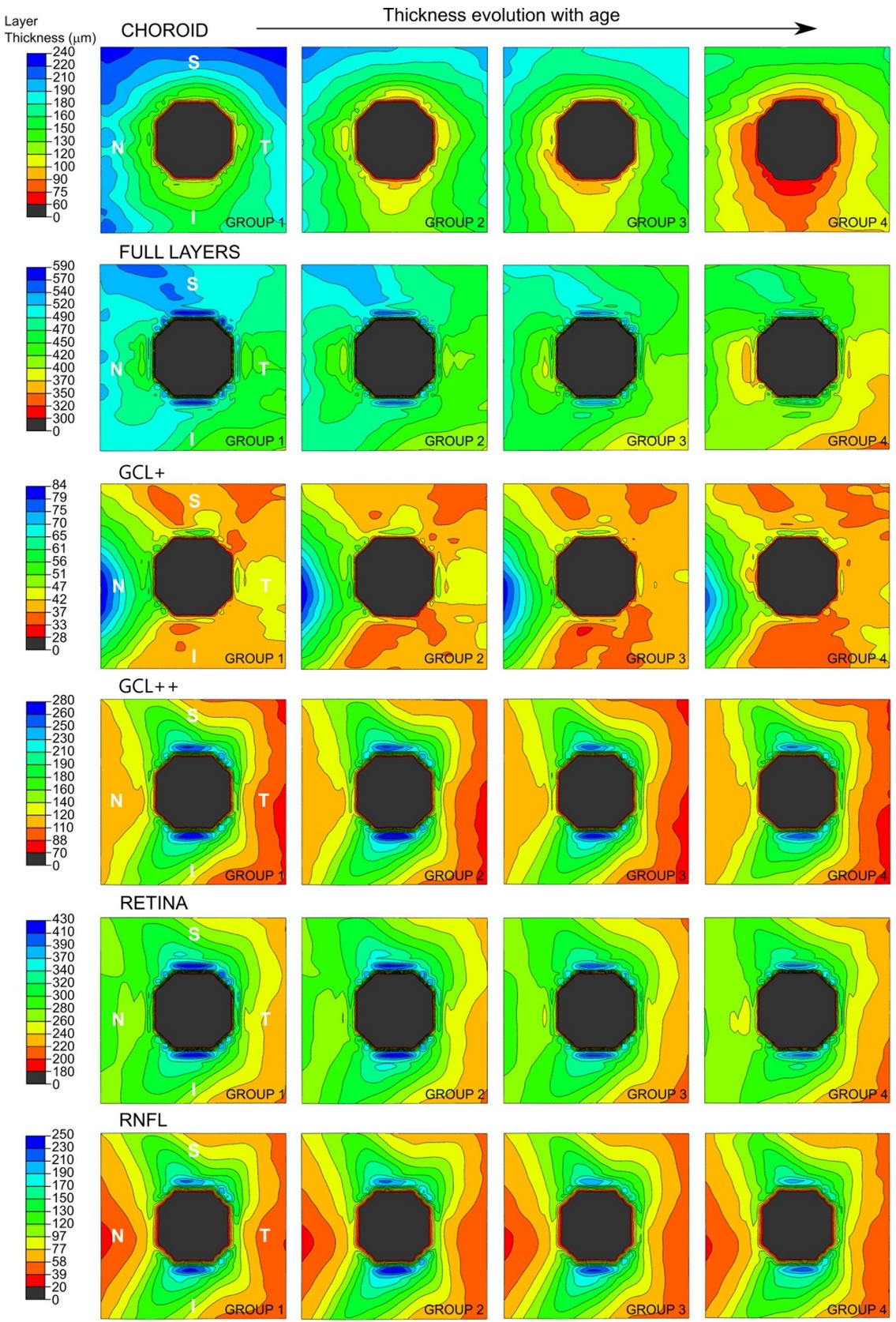

**Fig 4. Finite element contour map of the evolution of the thickness of each layer measured by Triton optical coherence tomography for the four age groups (20–34 years, 35–49 years, 50–64 years, and 65–80 years).** On the left, the contour bar represents the equivalence between colour and thickness value (in μm) for each layer. Blue zones correspond to thicker areas and red zones correspond to thinner ones. The optic nerve is represented in grey colour. Different scales have been used to represent the thickness value for each layer. It can be seen a progressive thinning of all layers with age, especially from 50 years onwards.

## Retina (from RNFL to photoreceptors layer)

There are no significant differences when comparing group 1 with group 2, but when comparing with group 3 significant thinning is found at TS and NS quadrants (p = 0.003 and p = 0.001) and superior and nasal halves (p = 0.001 and p = 0.006). Furthermore, when comparing with group 4, significant thinning is found in all quadrants and halves (p<0.001) except for NI quadrant (p = 0.041) (Table 2).

By contrast comparing group 2 with group 3, there seems to be a significant thinning at TS (p<0.001), NS (p = 0.013) and NI (p = 0.009) quadrants and all halves (p<0.001, p = 0.024, p = 0.008, p = 0.003), but when comparing with group 4 significant thinning occurs in all quadrants (p<0.001, p<0.001, p<0.001 and p = 0.008) and all halves (p<0.001). Finally, when

**Table 1. Mean ± standard deviation of four quadrants and halves for age groups in the retinal nerve fiber layer, GCL+ and GCL++, and comparison of thickness between age groups.**

|  |  | Group 1 (20–34 years) | Group 2 (35–49 years) | Group 3 (50–64 years) | Group 4 (65–79 years) | P |
|---|---|---|---|---|---|---|
| RNFL | TS Quadrant | 97.10±10.17 | 103.01±10.05 | 97.53±11.79 | 96.36±11.70 | 0.001* |
|  | TI Quadrant | 93.16±11.55 | 97.63±9.57 | 97.12±13.62 | 90.49±12.03 | <0.001* |
|  | NS Quadrant | 88.89±12.20 | 88.70±10.66 | 87.13±14.22 | 85.19±12.04 | 0.222 |
|  | NI Quadrant | 77.72±10.44 | 82.00±7.50 | 80.94±11.39 | 84.49±15.81 | 0.003* |
|  | Superior Half | 92.99±6.39 | 95.87±8.03 | 92.33±10.55 | 90.78±9.94 | 0.009 |
|  | Inferior Half | 85.44±6.15 | 89.81±6.08 | 89.03±9.17 | 97.49±10.30 | 0.007* |
|  | Nasal Half | 83.30±9.31 | 85.35±8.02 | 84.03±11.82 | 84.84±13.16 | 0.687 |
|  | Temporal Half | 95.13±9.60 | 100.32±7.41 | 97.33±11.89 | 93.42±10.73 | <0.001* |
| GCL + | TS Quadrant | 48.87±4.26 | 47.70±5.29 | 47.01±4.69 | 45.10±4.35 | <0.001* |
|  | TI Quadrant | 53.07±6.21 | 52.36±6.07 | 51.11±5.51 | 49.70±5.63 | 0.001* |
|  | NS Quadrant | 40.42±2.99 | 39.27±3.22 | 39.44±4.31 | 38.40±3.82 | 0.011 |
|  | NI Quadrant | 41.06±3.84 | 39.69±3.67 | 39.13±3.93 | 37.79±5.42 | <0.001* |
|  | Superior Half | 44.65±3.05 | 43.49±4.3 | 43.23±3.76 | 41.75±3.38 | <0.001* |
|  | Inferior Half | 47.06±4.14 | 46.03±4.14 | 45.15±3.85 | 43.75±4.53 | <0.001* |
|  | Nasal Half | 40.47±3.15 | 38.48±3.17 | 39.29±3.72 | 38.09±3.87 | <0.001* |
|  | Temporal Half | 50.97±4.82 | 50.03±5.44 | 49.06±4.74 | 47.40±4.56 | <0.001* |
| GCL++ | TS Quadrant | 145.98±9.61 | 150.72±11.23 | 144.55±11.72 | 141.47±13.00 | <0.001* |
|  | TI Quadrant | 146.23±9.25 | 150.01±10.43 | 148.25±13.14 | 140.20±13.29 | <0.001* |
|  | NS Quadrant | 129.28±13.08 | 128.00±11.89 | 126.49±15.07 | 123.45±13.76 | 0.046 |
|  | NI Quadrant | 118.78±10.08 | 121.70±8.98 | 120.01±11.65 | 122.10±16.72 | 0.263 |
|  | Superior Half | 137.63±8.03 | 139.36±10.03 | 135.52±11.74 | 132.46±12.03 | 0.001* |
|  | Inferior Half | 132.51±6.73 | 135.85±8.45 | 134.13±10.16 | 131.15±12.68 | 0.015 |
|  | Nasal Half | 124.03±10.07 | 124.85±9.62 | 123.25±12.53 | 122.78±14.55 | 0.709 |
|  | Temporal Half | 146.10±8.40 | 150.36±9.29 | 146.40±11.79 | 140.83±12.39 | <0.001* |

The asterisks show significant levels that overcome Bonferroni correction for multiple comparisons, Abbreviations: TS, temporal superior; TI, temporal inferior; NS, nasal superior; NI, nasal inferior.

**Table 2. Mean ± standard deviation of four quadrants and halves for age groups in the full layers, retina and choroid measurement, and comparison of thickness between age groups.**

|  |  | Group 1 (20–34 years) | Group 2 (35–49 years) | Group 3 (50–64 years) | Group 4 (65–79 years) | P |
|---|---|---|---|---|---|---|
| Full layers | TS Quadrant | 498.67±40.41 | 517.52±71.87 | 485.86±79.41 | 458.84±52.42 | <0.001* |
|  | TI Quadrant | 470.53±50.57 | 493.19±73.98 | 463.30±78.88 | 418.36±47.21 | <0.001* |
|  | NS Quadrant | 479.50±43.65 | 480.96±68.32 | 465.49±89.01 | 426.43±54.78 | <0.001* |
|  | NI Quadrant | 436.80±35.97 | 449.07±68.93 | 427.20±82.69 | 389.57±45.18 | <0.001* |
|  | Superior Half | 489.09±39.43 | 499.24±68.04 | 475.68±83.03 | 442.63±51.55 | <0.001* |
|  | Inferior Half | 453.67±40.77 | 471.13±69.78 | 445.25±79.36 | 403.97±44.68 | <0.001* |
|  | Nasal Half | 458.15±38.27 | 465.02±67.74 | 446.35±85.07 | 408.00±47.52 | <0.001* |
|  | Temporal Half | 484.60±44.81 | 505.35±71.76 | 474.58±78.35 | 438.60±47.09 | <0.001* |
| Retina | TS Quadrant | 300.15±9.62 | 303.03±13.05 | 294.89±14.05 | 290.65±10.68 | <0.001* |
|  | TI Quadrant | 293.30±10.88 | 298.22±14.59 | 294.78±15.79 | 285.90±13.29 | <0.001* |
|  | NS Quadrant | 274.31±12.44 | 272.71±14.01 | 266.56±18.82 | 262.47±13.48 | <0.001* |
|  | NI Quadrant | 259.40±8.85 | 261.13±10.94 | 255.70±16.01 | 255.04±16.13 | 0.012 |
|  | Superior Half | 287.23±9.01 | 287.87±12.24 | 280.73±15.29 | 276.56±11.16 | <0.001* |
|  | Inferior Half | 277.85±8.47 | 279.68±12.06 | 275.24±14.68 | 270.47±13.13 | <0.001* |
|  | Nasal Half | 266.85±9.31 | 266.92±11.81 | 261.13±16.75 | 258.76±14.06 | <0.001* |
|  | Temporal Half | 298.22±9.38 | 300.63±12.80 | 294.84±14.42 | 288.27±11.30 | <0.001* |
| Choroid | TS Quadrant | 173.28±40.68 | 203.81±77.25 | 168.16±74.45 | 145.42±48.17 | <0.001* |
|  | TI Quadrant | 149.75±46.90 | 179.46±72.31 | 145.66±73.86 | 109.60±39.44 | <0.001* |
|  | NS Quadrant | 175.66±43.38 | 193.17±68.84 | 174.84±82.28 | 140.19±47.74 | <0.001* |
|  | NI Quadrant | 147.86±34.92 | 169.42±68.43 | 147.11±77.33 | 110.51±35.24 | <0.001* |
|  | Superior Half | 174.47±39.34 | 198.49±71.18 | 146.39±77.27 | 142.81±45.52 | <0.001* |
|  | Inferior Half | 148.80±38.36 | 174.44±68.99 | 146.39±74.44 | 110.05±35.90 | <0.001* |
|  | Nasal Half | 161.76±38.04 | 181.30±67.82 | 160.98±78.97 | 125.35±38.98 | <0.001* |
|  | Temporal Half | 161.52±43.20 | 191.64±73.73 | 156.91±73.27 | 127.51±41.66 | <0.001* |

The asterisks show significant levels that overcome Bonferroni correction for multiple comparisons. Abbreviations: TS, temporal superior; TI, temporal inferior; NS, nasal superior; NI, nasal inferior.

comparing group 3 with group 4 significant thinning is found at TS (p = 0.008) and TI (p<0.001) quadrants and superior (p = 0.016) and temporal (p<0.001) halves.

## Choroid analysis

Choroid howed significant differences between age groups for all quadrants and halves (Table 2). At second age group (35 to 49 years) it seems to exist a choroidal thickening at superior half (p = 0.001) when compared with group 1. By contrast, group 3 presents a thinner choroidal layer compared with previous groups, but this thinning is just significant when compared with group 2 for TS and TI quadrants, and for temporal half (p<0.001). The thinnest choroidal values can be seen for group 4, which is significant when comparing with group 1, 2 of 3 for every parameter analyzed (p<0.001).

Except the GCL+ with a linear thinning ratio of -0.050 μm/year, the tendency in the other layers differs. A thickening ratio of 0.476 μm/year for the full layers complex, 0.010 μm/year for the retina, 0.856 μm/year for the choroid, 0.125 μm/years for the RNFL and 0.087 μm/years for the GCL++ is observed until de third decade of life. Subsequently a thinning ratio of -1.406 μm/year for the full layers complex, -0.058 μm/year for the retina, -1.364 μm/year for the choroid, -0.084 μm/years for the RNFL and -0.131 μm/years for the GCL++ is observed from de third decade of life.

## Discussion

Previous histopathological studies of the choroid have shown that choroidal thickness decreases with age; this finding has been confirmed by posterior in-vivo studies [7, 14, 17, 21]. Choroidal thinning on OCT has been associated with ophthalmological conditions such as age-related macular degeneration, age-related choroidal atrophy, axial length, or systemic conditions such as diabetes, Alzheimer's disease, cognitive impairment, vascular diseases or obesity [11, 21, 22].

In addition, the loss of vascular perfusion related to age has been described in previous studies performed with OCT-Angiography [23], even this alteration of blood flow could be observed by magnetic resonance [24]. Furthermore, a significant reduction of retrobulbar circulation related to age has been found in Doppler studies, this reduction could be also related with the increased prevalence of cardiovascular risks related with age such as hypertension, diabetes, lipid disorders and sedentary lifestyle [25–28].

Traditionally, the age of 40 years has been established as the age of onset of the classic ophthalmological alterations, although other pathologies such as macular alterations are more common beyond 50 years, so these changes in the choroid can be a predictive factor of future clinical alterations [7, 29, 30].

In our study, we have seen that global retinal tendency is toward thickening in the third decade of life (group 2) and towards thinning after the fourth decade (group 3). Regarding the choroid, the same pattern was seen and also for the rest of layers, except at GCL+ layer, where there is a continuous thinning from group 1 to group 4. These results are consistent with those of Hanumunthadu et al. 2018, their age-stratified analysis suggested that choroidal thickness was smaller in children and younger adult population, and it appeared to decrease again in older adults [21].

In contrast with published studies, our analysis has focused on a division by quadrants and halves in the peripapillary area. We observed that superior half and quadrants are thicker than their inferior homologous, however, it seems that there is no difference when comparing nasal and temporal orientation. As we have seen previously, group 1 seems to have a very similar choroidal thickness to group 3 in practically all the quadrants and halves, except in the superior half, where apparently the thickness of the choroid is greater in this group.

Comparing the thinning of the total retina and the one produced specifically in the choroid, we can see that it is in group 4 where the thinning of this last layer is more pronounced, especially in the inferior half of the retina. Same thing happens for groups 1 and 2, but not in 3, where thinning is symmetric when comparing the superior quadrants and halves with the inferior ones and the nasal quadrants and halves with the temporal ones.

The potential clinical application of our findings is to know which thickness values in each layer should be considered pathological. This study may be the base to build a normative database that allows health personnel to improve the interpretation of retinal measurements, even with a color scale similar to that one used by most OCT devices (red color means pathological thinning, yellow color indicated slight thinning, green color means normal, blue indicates slight thickening and purple means abnormal thickening); this helps to determine which layers are out of normal limits in each range of age. This is an important limitation of the Triton current software because we can visualize numerous measurements of all retinal layers, but ophthalmologists do not know if these values are normal for the age of each subject or patient.

Our findings suggest that there is a progressive and physiological thinning of all retinal layers from the third decade of life. In any case, it would be useful to extend this study further with a larger population of ordinary individuals across a range of different ethnicities and from longer geographic areas, to clarify the physio-pathological mechanisms that affect retinal layers.

## Supporting information

**S1 Data.**

(SAV)

## Author Contributions

**Data curation:** Elisa Viladés, Elvira Orduna.

**Formal analysis:** Amaya Pérez-del Palomar, José Cegoñino, Elena Garcia-Martin.

**Investigation:** Elisa Viladés.

**Resources:** Elvira Orduna.

**Software:** Amaya Pérez-del Palomar, José Cegoñino.

**Supervision:** Javier Obis, María Satue, Luis E. Pablo.

**Writing – original draft:** Elisa Viladés.

**Writing – review & editing:** Javier Obis, María Satue, Elvira Orduna, Luis E. Pablo, Marta Ciprés, Elena Garcia-Martin.

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
