## [Decision Letter · Decision Letter 0]

1 Jun 2020

PONE-D-20-02557

Physiological changes in retinal layers thicknesses measured with swept source optical coherence tomography.

PLOS ONE

Dear Dr. Garcia-Martin,

Thank you for submitting your manuscript to PLOS ONE. After careful consideration, we feel that it has merit but does not fully meet PLOS ONE’s publication criteria as it currently stands. Therefore, we invite you to submit a revised version of the manuscript that addresses the points raised during the review process.

Reviewers have provided constructive criticisms that need to be addressed satisfactorily during revision. The reviewers also opin that novel information of intraretinal layers over aging has not been provided in the manuscript. 

We look forward to receiving your revised manuscript.

Kind regards,

Sanjoy Bhattacharya

Academic Editor

PLOS ONE

Journal Requirements:

2. Thank you for including your ethics statement:  "All subjects gave detailed consent to participate in this study, which was conducted in accordance with the guidelines determined by the Ethics Committee of the Miguel Servet Hospital and the principles of the Declaration of Helsinki."

Please amend your current ethics statement to confirm that your named institutional review board or ethics committee specifically approved this study.

3. Please include a copy of Table 4 which you refer to in your text on page 9.

Reviewers' comments:

Reviewer's Responses to Questions

**Comments to the Author**

1. Is the manuscript technically sound, and do the data support the conclusions?

Reviewer #1: Partly

Reviewer #2: Yes

2. Has the statistical analysis been performed appropriately and rigorously? 

Reviewer #1: I Don't Know

Reviewer #2: I Don't Know

3. Have the authors made all data underlying the findings in their manuscript fully available?

Reviewer #1: Yes

Reviewer #2: Yes

4. Is the manuscript presented in an intelligible fashion and written in standard English?

Reviewer #1: No

Reviewer #2: Yes

5. Review Comments to the Author

Reviewer #1: The authors collects SS-OCT scans and analyzed the results from a large cohort which is interesting. However, the paper did not provide novel information of intraretinal layers over aging.

1. the authors did not present the results of all intraretinal layers such as INL, OPL, ONL and PR. However, some selected layers and their combinations are presented and the majority of these presented layers are not commonly used in clinic. For example, we all use RNFL and GCIPL in addition to the retinal thickness.

2. Although the SS-OCT scans a large area which may be better than the 6 x 6 mm scan area commonly used in clinic and research. On the other hand, the large area may prevent further comparison if the normality data is not available.

3. The authors did not provide the details of scan settings such scan lines and area, and did not provide the details of 3D segmentation methods.

4. The authors will need to provide a cross-sectional OCT image and show the segmentation boundaries and 3D map.

5. The age grouping is highly arbitrary and the authors need to provide the justification of the cut-off setting. It would be a good approach to use scatterplot to show the trend.

5. The changes of intraretinal layers have been well documented and this manuscript did not report new findings, nor new database (since the device is not commonly used in the clinic).

6. The numbers appear to be wrong since the retinal thickness cannot be about 500,00 microns. It should be about 500.00 microns. Similarily, P should be < 0.001, instead of P < 0,001.

7. The simple conclusion is the thinning of retinal layer, which is well documented. However, the authors did not provide the thinning rate.

8. There are missing data in the center as shown in Fig. 3. Why is the center area trimmed and shown black. Did the center of the fovea was aligned from each map? Or this is just a numerical demonstration.

Reviewer #2: a thickening until de third decade of live,: I will change it to third decade of life not live.

line 75: possible not posible

line 84: systemic conditions like diabetes mellitus 11 or migraine: why considering migraine systemic condition?

Line 91: advent: I will write it advantage instead

line 93: Even so,: should be even though

did U mention in this article the race of the patients, like white, AA or east Asian...etc, I mean U did not mention the ethnicity of your study population

6. PLOS authors have the option to publish the peer review history of their article (what does this mean?). If published, this will include your full peer review and any attached files.

Reviewer #1: No

Reviewer #2: Yes: Feras Mohder

---

## [Author Response · Author response to Decision Letter 0]

6 Jul 2020

We thank reviewers and editor for providing us with very pertinent and helpful comments

---

## [Decision Letter · Decision Letter 1]

28 Sep 2020

Physiological changes in retinal layers thicknesses measured with swept source optical coherence tomography.

PONE-D-20-02557R1

Dear Dr. Garcia-Martin,

We’re pleased to inform you that your manuscript has been judged scientifically suitable for publication and will be formally accepted for publication once it meets all outstanding technical requirements.

Kind regards,

Sanjoy Bhattacharya

Academic Editor

PLOS ONE

Additional Editor Comments (optional):

Reviewers' comments:

Reviewer's Responses to Questions

**Comments to the Author**

1. If the authors have adequately addressed your comments raised in a previous round of review and you feel that this manuscript is now acceptable for publication, you may indicate that here to bypass the “Comments to the Author” section, enter your conflict of interest statement in the “Confidential to Editor” section, and submit your "Accept" recommendation.

Reviewer #1: All comments have been addressed

Reviewer #2: All comments have been addressed

2. Is the manuscript technically sound, and do the data support the conclusions?

Reviewer #1: Yes

Reviewer #2: Yes

3. Has the statistical analysis been performed appropriately and rigorously? 

Reviewer #1: Yes

Reviewer #2: Yes

4. Have the authors made all data underlying the findings in their manuscript fully available?

Reviewer #1: Yes

Reviewer #2: Yes

5. Is the manuscript presented in an intelligible fashion and written in standard English?

Reviewer #1: Yes

Reviewer #2: Yes

6. Review Comments to the Author

Reviewer #1: This reviewer has no further comments. All comments in the previous review have been addressed and changes have been made.

Reviewer #2: What are the reasons that the groups in your study are not in equal numbers?

in line 38: u wrote double in , you may delete one of them.

in line 41: U can replace de with the.

in line 70: " from retina thanks to a wavelength": I really did not understand what do U mean in this line?

7. PLOS authors have the option to publish the peer review history of their article (what does this mean?). If published, this will include your full peer review and any attached files.

Reviewer #1: No

Reviewer #2: **Yes: **Feras Mohder

---

## [Editor Report · Acceptance letter]

5 Oct 2020

PONE-D-20-02557R1 

Physiological changes in retinal layers thicknesses measured with swept source optical coherence tomography. 

Dear Dr. Garcia-Martin:

I'm pleased to inform you that your manuscript has been deemed suitable for publication in PLOS ONE. Congratulations! Your manuscript is now with our production department. 

Kind regards, 

on behalf of

Dr. Sanjoy Bhattacharya 

Academic Editor

PLOS ONE